# Effect of Short-Term Restraint Stress on the Expression of Genes Associated with the Response to Oxidative Stress in the Hypothalamus of Hypertensive ISIAH and Normotensive WAG Rats

**DOI:** 10.3390/antiox13111302

**Published:** 2024-10-26

**Authors:** Yulia V. Makovka, Dmitry Yu. Oshchepkov, Larisa A. Fedoseeva, Arcady L. Markel, Olga E. Redina

**Affiliations:** 1Federal Research Center Institute of Cytology and Genetics, Siberian Branch of Russian Academy of Sciences, 630090 Novosibirsk, Russia; makovkayv@bionet.nsc.ru (Y.V.M.); diman@bionet.nsc.ru (D.Y.O.); fedoseeva@bionet.nsc.ru (L.A.F.); markel@bionet.nsc.ru (A.L.M.); 2Department of Natural Sciences, Novosibirsk State University, 630090 Novosibirsk, Russia; 3Kurchatov Genomic Center Institute of Cytology and Genetics, Siberian Branch of Russian Academy of Sciences, 630090 Novosibirsk, Russia

**Keywords:** hypothalamus, single short-term restraint stress, gene expression, oxidative phosphorylation, response to oxidative stress, ISIAH hypertensive rat strain

## Abstract

Normotensive and hypertensive organisms respond differently to stress factors; however, the features of the central molecular genetic mechanisms underlying the reaction of the hypertensive organism to stress have not been fully established. In this study, we examined the transcriptome profiles of the hypothalamus of hypertensive ISIAH rats, modeling a stress-sensitive form of arterial hypertension, and normotensive WAG rats at rest and after exposure to a single short-term restraint stress. It was shown that oxidative phosphorylation is the most significantly enriched process among metabolic changes in the hypothalamus of rats of both strains when exposed to a single short-term restraint stress. The analysis revealed DEGs representing both a common response to oxidative stress for both rat strains and a strain-specific response to oxidative stress for hypertensive ISIAH rats. Among the genes of the common response to oxidative stress, the most significant changes in the transcription level were observed in *Nos1*, *Ppargc1a*, *Abcc1*, *Srxn1*, *Cryab*, *Hspb1*, and *Fosl1*, among which *Abcc1* and *Nos1* are associated with hypertension, and *Fosl1* and *Ppargc1a* encode transcription factors. The response to oxidative stress specific to hypertensive rats is associated with the activation of the *Fos* gene. The DEG’s promoter region enrichment analysis allowed us to hypothesize that the response to oxidative stress may be mediated by the participation of the transcription factor CREB1 (Cyclic AMP-responsive element-binding protein 1) and the glucocorticoid receptor (NR3C1) under restraint stress in the hypothalamus of both rat strains. The results of the study revealed common and strain-specific features in the molecular mechanisms associated with oxidative phosphorylation and oxidative stress response in the hypothalamus of hypertensive ISIAH and normotensive WAG rats following a single short-term restraint stress. The obtained results expand the understanding of the most significant molecular targets for further research aimed at developing new therapeutic strategies for the prevention of the consequences of acute emotional stress, taking into account the hypertensive state of the patient.

## 1. Introduction

Arterial hypertension is one of the significant risk factors predisposing to fatal complications in cardiovascular diseases. The mechanisms underlying the pathological processes leading to the development of hypertension have not been fully elucidated. There is much evidence that processes associated with oxidative stress and chronic inflammation play a significant role in the pathogenesis of hypertension, leading to endothelial damage, the development of vascular rigidity, and, accordingly, changes in vascular tone and blood flow resistance [1,2,3,4]. There is currently no doubt that the central nervous system plays a crucial role in the pathogenesis of hypertension [5]. Oxidative stress in brain structures contributes to the development of hypertension through the activation of the sympathetic nervous system [5,6,7]. A significant role of oxidative stress has been shown in the brainstem [7,8,9], the hypothalamus [10,11], as well as in other parts of the brain [12]. It has been shown that normotensive and hypertensive organisms react differently to the effects of stress factors [13,14]. The complexity of the mechanisms of oxidative stress action on the development of the stress-sensitive form of arterial hypertension and the insufficiently complete study of the physiological and molecular-genetic mechanisms that determine the characteristics of the body’s reaction to stress in hypertensive patients necessitate research in this area in order to identify potentially significant targets for use in medical practice [15].

ISIAH (inherited stress-induced arterial hypertension) rats were selected for a sharp increase in systolic blood pressure under conditions of short-term (30 min) restraint stress [16,17]. ISIAH rats are characterized as a model of stress-sensitive hypertension with genetically determined activation of the hypothalamic–pituitary–adrenal and sympathetic adrenal systems [18]. A comparison of gene expression profiles of the brainstem, hypothalamus, adrenal glands, and kidneys of hypertensive ISIAH rats and control normotensive Wistar Albino Glaxo (WAG) rats revealed the presence of significant interstrain differences in the transcription levels of numerous genes associated with hypertension and blood pressure regulation, as well as with the functioning of the central nervous system [19,20,21,22,23]. The obtained results allowed us to suggest the presence of a genetically determined state of functional stress in the analyzed brain regions and peripheral target organs in hypertensive ISIAH rats compared to normotensive WAG rats. The study of changes in the hypothalamic transcriptomes of rats of these strains in response to a single short-term (2 h) restraint stress allowed us to identify a group of genes representing a common response to stress for both rat strains, as well as genes representing a strain-specific response [14]. In this article [14], the main attention is paid to groups of genes associated with ion transport, response to endogenous stimuli, and signaling. The current study provides a detailed overview of the common and strain-specific features in the molecular mechanisms associated with oxidative phosphorylation and oxidative stress response in the hypothalamus of ISIAH and WAG rats following a single short-term restriction of their mobility and discusses the possible role of these molecular events in the regulation of blood pressure levels in hypertensive and normotensive rats.

## 2. Materials and Methods

### 2.1. Animals

The experiment was conducted on three-month-old male hypertensive rats of the ISIAH/Icgn (Inherited Stress-Inducible Arterial Hypertension) strain and normotensive rats of the WAG/GSto-Icgn (Wistar Albino Glaxo) strain. The work was carried out on the basis of the conventional vivarium of the Center for Genetic Resources of Laboratory Animals of the Institute of Cytology and Genetics of the Siberian Branch of the Russian Academy of Sciences, where the animals were kept in standard conditions, without restrictions in water and balanced feed.

To analyze the hypothalamic transcriptomes using the RNA-Seq method, 4 groups of 7 animals each were formed: (1) ISIAH_control; (2) WAG_control; (3) ISIAH_stress; (4) WAG_stress. The time course of the experiment is shown in Figure 1. All rats had their basal systolic blood pressure (BP) measured using the tail-cuff method under light ether anesthesia to prevent the influence of the rat’s emotional reaction on the BP measurements [18]. Hypothalamic samples were collected 7 days after basal BP measurement. For two experimental groups, ISIAH_stress and WAG_stress, hypothalamic sample collection was performed immediately after a 2 h restraint (emotional) stress. Restriction was carried out by placing the rat in a tight wire-mesh cage for 2 h. This type of stress limits the rat’s locomotor activity without causing pain. BP levels after restraint stress were measured without anesthesia. The details of this procedure have been described previously [14]. After completion of the BP measurement, immediate decapitation, isolation of the hypothalamus (according to known anatomical location [24]), and homogenization in 700 μL of ExtractRNA reagent (Eurogen, Moscow, Russia) using 500 μL of Lysing matrix D (Cat#6540434 MP Biomedicals, Solon, OH, USA) for 20 s at 18,000 rpm in a Super FastPrep-2 homogenizer (MP Biomedicals, Solon, OH, USA) were performed. Plasma corticosterone concentrations were measured using the Corticosterone ELISA Kit (Ab108821, Abcam, Boston, MA, USA). Changes in BP and plasma corticosterone concentrations in ISIAH and WAG rats upon exposure to stress are shown in [14].

The experiment was carried out in accordance with the International Rules for Conducting Work Using Laboratory Animals and approved by the Bioethics Committee of the Institute of Cytology and Genetics SB RAS (Novosibirsk, Russia), protocol No. 115 dated 20 December 2021.

### 2.2. RNA-Seq Analysis

RNA isolation was carried out at the Institute of Genomic Analysis (Moscow, Russia). Sample preparation and hypothalamic transcriptome sequencing according to the manufacturer’s protocols (MGI Tech Co., Ltd., Shenzhen, China) were performed at the BGI Hongkong Tech Solution NGS laboratory (Hong Kong, China). Paired-end sequencing of cDNA libraries was performed using the DNBSEQ platform (DNBSEQ Technology, Hong Kong, China) with a read length of 150 bp and a sequencing depth of over 30 million uniquely mapped reads. All samples were analyzed as biological replicates.

After assessing the quality of the obtained sequencing data using the FastQC program (version 0.11.5 [25]), a total of 1,267,436,623 (98.45%) nucleotide reads were mapped to the reference rat genome mRatBN7.2/rn7 (rn7 assembly Wellcome Sanger Institute November, 2020) using the STAR software package, version 2.7.10a [26]. Statistical analysis and differential gene expression calculation were performed in DESeq2 v1.30.1 [27]. To account for unwanted variation in the data caused by possible random systematic bias in the sample preparation process, we applied surrogate variable analysis (SVA) [28]. Significant surrogate variables were further included as factors in the differential expression analysis in DESeq2 according to the documentation. All genes demonstrating a sufficient expression level above the threshold (the sum of gene coverages for all libraries is more than 10 reads) were included in the differential expression analysis. Differentially expressed genes were determined taking into account the correction for multiple comparisons. The significance threshold corresponded to the adjusted *p*-value < 5%.

Comparative analysis of differential gene expression was performed between the groups ISIAH_stress-ISIAH_control (the list of resulting DEGs is designated as ISIAH_DEGs) and WAG_stress-WAG_control (the list of resulting DEGs is designated as WAG_DEGs).

### 2.3. Functional Annotation of DEGs

For functional annotation of DEGs, the Rat Genome Database [29] and DAVID (The Database for Annotation, Visualization and Integrated Discovery), which includes analysis in the Gene Ontology and KEGG databases, were used [30,31]. The functional enrichment networks were constructed by means of the STRING database [32]. The enrichment analysis of gene promoter regions with transcription factor binding sites was performed using the Enrichr resource [33]. An atlas of combinatorial transcriptional regulation in mice and humans [34] was utilized for the identification of DEG encoding transcription factors. The results of functional annotation were visualized using the SRplot, a free online platform for data visualization and graphing [35].

### 2.4. Statistical Methods

RNA-Seq data (in FPKM values) were log2-transformed, centered, normalized, and scaled using principal coordinate analysis based on Euclidean distance metrics, followed by PLS-DA analysis [36]. These procedures resulted in the construction of PLS-DA axes that maximized the distances between stressed and control rats. Then, the calculation of Pearson correlation coefficients helped to find the set of variables (expressed genes) that were expected to maximize the covariance between a fixed dummy matrix representing group membership (for stressed and control rats) and gene expression in these animals. The calculated correlation revealed the genes characterized by the greatest deviation along the first functionally significant synthetic axis (PLS-DA axis 1). These genes were hypothesized to be candidates for contributing most to intergroup differences. Normalized RNA-Seq data were also used for Pearson correlation analysis. Pearson correlation coefficient critical value of 0.661 [df = 12; two-tailed *p* value = 0.01] was considered significant. The software packages STATISTICA 12.0 (StatSoft, Tulsa, OK, USA) and JACOBI, version 4.3.21 (Novosibirsk, Russia) [37] were used for data analysis and presentation.

## 3. Results

### 3.1. Characteristics of DEGs Detected in the Hypothalamus of ISIAH and WAG Rats Exposed to Single Restraint Stress for 2 h

The genes that significantly changed their transcription level in the hypothalamus of hypertensive ISIAH rats (ISIAH stress-ISIAH control) and normotensive WAG rats (WAG stress-WAG control) under the influence of a single short-term (2 h) restraint stress were identified. The transcription level during stress was changed by 3603 hypothalamic genes in hypertensive ISIAH rats (ISIAH DEGs) and 3157 genes in WAG rats (WAG DEGs). Among them, 1541 DEGs are common to both rat strains (Figure 2a). Accordingly, it can be concluded that a large number of DEGs are involved in both the common and strain-specific responses to stress. Both activation and decrease in the transcription level of a large number of DEGs are observed in both rat strains (Figure 2b).

### 3.2. Functional Annotation of DEGs

#### 3.2.1. Identification of the Most Enriched KEGG Pathways

Functional annotation of DEGs allowed us to identify the most significantly enriched metabolic pathways and associated pathological processes, the functioning of which can be altered by a single short-term (2 h) restraint stress in the hypothalamus of hypertensive ISIAH and normotensive WAG rats (Appendix A). Some metabolic pathways were identified as significantly enriched in the analysis of DEGs of both rat strains (marked with an asterisk in Appendix A).

To identify DEGs that contribute the most to intergroup differences, we used the PLS-DA approach (Appendix A), followed by correlation analysis between the coordinates of objects (rats) along the first PLS-DA axis and the level of gene transcription measured in the rat hypothalamus at rest and after exposure to restraint stress (Appendix A).

#### 3.2.2. Identification of DEGs and Metabolic Pathways That Contribute Most to Intergroup Differences

A comparison of ISIAH_stress and ISIAH_control rat groups revealed 1219 DEGs, and a comparison of WAG_stress and WAG_control rat groups revealed 161 DEGs contributing most significantly (r > 0.90) to the intergroup differences. Functional annotation of these DEGs (Figure 3) shows that oxidative phosphorylation is the most significant process among the metabolic changes in the hypothalamic gene function profile of both rat strains exposed to single short-term restraint stress.

#### 3.2.3. Characteristics of DEGs for Oxidative Phosphorylation

A total of 74 DEGs in the hypothalamus of ISIAH rats and 40 DEGs in the hypothalamus of WAG rats were assigned to the Oxidative phosphorylation pathway (Appendix A). Lists of these DEGs are presented in Appendix A. The expression of almost all of them (except for one gene in ISIAH rats and two genes in WAG rats) increased upon restraint stress (Figure 4), suggesting activation of the Oxidative phosphorylation pathway in both rat strains. A comparison of the lists of DEGs associated with the term Oxidative phosphorylation in ISIAH and WAG rats revealed that 34 DEGs are common (Figure 4). In the hypothalamus of hypertensive rats, a significant number of DEGs associated with oxidative phosphorylation were observed, which can be considered as an ISIAH-strain-specific response to stress.

Functional analysis of 34 common and 40 ISIAH-specific DEGs associated with oxidative phosphorylation was performed to identify possible strain-specific features of the oxidative phosphorylation process in the hypothalamus of hypertensive ISIAH rats. The analysis showed that the common and ISIAH-specific responses are described by the same terms of biological processes (Mitochondrial ATP synthesis coupled electron transport, Mitochondrial respiratory chain complex I assembly) and molecular functions (Electron transfer activity, Proton transmembrane transporter activity) (Figure 5). Accordingly, activation of the oxidative phosphorylation metabolic pathway in the hypothalamus of rats of both strains differs only in the number of involved DEGs. This observation, together with the fact that almost all (28 out of 34) common DEGs had higher Log2FC values in ISIAH rats compared to WAG (Figure 4), suggests that a stronger activation of the oxidative phosphorylation pathway by stress is observed in the hypothalamus of hypertensive ISIAH rats than in the hypothalamus of normotensive WAG rats.

As can be seen from Appendix A, most of the DEGs involved in Oxidative phosphorylation form the basis of several metabolic pathways related to Pathways of neurodegeneration (Parkinson disease, Huntington disease, Alzheimer disease, Amyotrophic lateral sclerosis), as well as metabolic pathways associated with the development of socially significant diseases (Diabetic cardiomyopathy, Chemical carcinogenesis—reactive oxygen species, Non-alcoholic fatty liver disease, Prion disease), or with the maintenance of physiological functions (Thermogenesis, Retrograde endocannabinoid signaling). This observation emphasizes the functional significance of the change in the expression of genes associated with the Oxidative phosphorylation metabolic pathway.

To identify biological processes that are most significantly altered in response to a single short-term restraint stress, a functional analysis of DEGs in the Gene Ontology database was performed.

#### 3.2.4. Gene Ontology (GO) Analysis

Functional annotation of DEGs in the Gene Ontology database allowed us to identify biological processes that were most significantly altered in response to the short-term restraint stress used in the experiment. Eighteen terms common to the stress response in the hypothalamus of ISIAH and WAG rats were found (Figure 6). As can be seen, the identified gene groups are represented by similar characteristics in terms of gene numbers and *p* values.

Strain-specific GO terms characterizing groups of DEGs associated with the response to stress stimuli are presented in Figure 7. The most significantly enriched term specific to ISIAH rats is response to oxidative stress (170 DEGs). In WAG rats, the most significantly enriched term is Response to ischemia (33 DEGs).

### 3.3. Analysis of Response to Oxidative Stress

#### 3.3.1. DEGs Associated with Response to Oxidative Stress

A list of 170 genes associated with the term “response to oxidative stress” that were identified in the hypothalamus of hypertensive ISIAH rats is presented in the Appendix A. The majority of these DEGs (109 DEGs, 64.1%) were upregulated by 2 h of restraint stress. According to the DEG annotation in the Rat Genome Database, 55 of the 170 genes were associated with hypertension (see Appendix A).

Functional annotation of hypothalamic DEGs in WAG rats did not reveal the term “response to oxidative stress”. However, of the 170 genes associated with response to oxidative stress in ISIAH rats, 59 genes were characterized as DEGs in the hypothalamus of WAG rats (Appendix A). Among them, 21 genes were associated with hypertension. All common 59 DEGs changed the transcription level unidirectionally in both rat strains (Figure 8). Most DEGs (41 DEGs, 69.5%) demonstrated a more significant change in the transcription level under stress in the hypothalamus of ISIAH rats.

Among the 59 common DEGs (Figure 8), several genes showed significant changes in transcription levels (more than 1.5-fold in at least one strain). Characterization of 111 genes that changed transcription levels in response to stress in the hypothalamus of hypertensive ISIAH rats only (Figure 9) shows that several of them also changed transcription levels quite significantly under stress.

#### 3.3.2. Characteristics of Oxidative-Stress-Response-Associated DEGs That Showed the Greatest Changes in Transcript Levels

The list of DEGs associated with the response to oxidative stress (Figure 8 and Figure 9) includes 11 DEGs characterized by a more than 1.5-fold change in transcription level under stress (Table 1). Seven of these genes statistically significantly change the transcription level in the hypothalamus of both rat strains, and four genes change the transcription level only in the hypothalamus of hypertensive ISIAH rats. The hypertension-associated genes *Nos1* and *Abcc1* are downregulated in the hypothalamus of both rat strains, and *Cyp1b1* and *Fos* are ISIAH-specific DEGs associated with hypertension.

The genes *Ppargc1a*, *Fosl1*, and *Fos* encode transcription factors. *Ppargc1a* and *Fosl1* change the transcription level under stress uniformly in both rat strains. Activation of the *Fos* gene is observed in the hypothalamus of hypertensive ISIAH rats only.

Considering that changes in the expression level of transcription factors can trigger cascade changes in the regulation of numerous physiological processes, it can be concluded that the general response to oxidative stress may be associated with the expression of the *Ppargc1a* and *Fosl1* genes, and the hypothalamic response to oxidative stress specific to hypertensive rats may be associated with the activation of the *Fos* gene.

Functional annotation of the 11 most highly altered DEGs showed that they are associated with such biological processes as response to epinephrine, response to temperature stimulus, response to cAMP, response to alcohol, response to cytokine, and response to various hormonal stimuli: response to glucocorticoid, response to progesterone, response to estradiol, and response to peptide hormone (Figure 10).

The results of the correlation analysis between these DEGs are shown in Figure 11. A highly reliable positive correlation was found between the expression of the *Abcc1*, *Nos1*, and *Ppargc1a* genes in the hypothalamus of both rat strains, and in the hypothalamus of ISIAH rats, the expression of these three genes also positively correlated with the transcription level of *Cyp1b1* and *Prkaa2*. The expression of the above-mentioned genes inversely correlated with the transcription level of *Cryab*. The most statistically significant positive correlations of *Fosl1* expression were found with the transcription level of the *Cryab* and *Srxn1* genes in the hypothalamus of both strains. Thus, the expression of three of the four DEGs associated with hypertension correlates with the expression of *Ppargc1a*, which encodes a transcription factor. ISIAH-specific activation of *Fos* gene expression, encoding a transcription factor associated with hypertension, most significantly correlates with *Hspb1* expression in the hypothalamus of ISIAH rats.

These DEGs can also be coregulated with other transcription factor genes associated with the response to oxidative stress. To obtain a more complete picture of coregulation, all transcription factor genes associated with the response to oxidative stress were identified.

#### 3.3.3. Characteristics of DEGs Encoding Transcription Factors Associated with Response to Oxidative Stress

The list of 170 DEGs associated with the response to oxidative stress in the hypothalamus of ISIAH rats contains 24 genes encoding transcription factors, most of which (15 DEGs) are activated (Table 2). Eight DEGs encoding transcription factors associated with the response to oxidative stress changed the transcription level in the hypothalamus of both rat strains. These changes were unidirectional.

Correlations of the expression of DEGs that most significantly change the transcription level under restraint stress (Table 1) with the expression of DEGs encoding transcription factors associated with the response to oxidative stress are presented in Table 3.

The greatest number of statistically highly reliable correlations in the hypothalamus of ISIAH rats were obtained for the expression of the *Abcc1*, *Cryab*, *Nos1*, *Ppargc1a*, *Cyp1b1*, and *Prkaa2* genes. Their transcription levels correlate with the transcription levels of several genes encoding transcription factors associated with the response to oxidative stress, including transcription factors associated with hypertension (*Arnt*, *Atf4*, *Ddit3*, *Foxo1*). In the hypothalamus of normotensive WAG rats, the most statistically significant correlations were found between the expression of the *Abcc1* and *Nos1* genes with the transcription factors *Arnt*, *Atm*, and *Ppargc1a*.

According to the obtained results, it can be assumed that genes associated with the response to oxidative stress, *Abcc1*, *Nos1*, and the *Arnt* gene encoding a transcription factor, can play a significant role in the regulation of blood pressure levels during stress, regardless of the hypertensive status of the body.

The ISIAH-specific response to stress associated with hypertension is represented by the *Cyp1b1* and *Fos* genes. The expression of *Cyp1b1* correlates with the expression of several transcription factor genes, in contrast to *Fos*, the expression of which correlates only with the expression of *Jun*.

#### 3.3.4. Enrichment Analysis of Promoter Regions of DEGs Associated with Response to Oxidative Stress by Transcription Factor Binding Sites

Using the Enrichr resource, an enrichment analysis of the promoter regions of DEGs associated with the response to oxidative stress was performed. This analysis revealed two terms associated with the binding sites of the transcription factor CREB1 (Cyclic AMP-responsive element-binding protein 1) and the glucocorticoid receptor NR3C1 (nuclear receptor subfamily 3, group C, member 1) (Table 4).

The results of the analysis showed that 38 DEGs can be regulated by the transcription factor CREB1, and NR3C1 can be involved in the regulation of the expression of 12 DEGs out of 170 associated with the response to oxidative stress in the hypothalamus of ISIAH rats. Three genes (*Bcl2l1*, *Hyal2*, and *Txnip*) are regulated by both CREB1 and NR3C1, and the remaining genes are regulated only by one of them. According to the RGD database, both genes encoding transcription factors CREB1 and NR3C1 are associated with hypertension. They can regulate the expression of 22 genes associated with hypertension among 170 DEGs associated with the response to oxidative stress. Unlike NR3C1, CREB1 can participate in the regulation of the transcription level of nine transcription factor genes differentially expressed in the hypothalamus of ISIAH rats and associated with the response to oxidative stress (Table 4).

Enrichr analysis showed that among 59 DEGs associated with response to oxidative stress in the hypothalamus of WAG rats, 16 genes could be regulated by CREB1, and 7 genes could be regulated by NR3C1 (Table 4). Two of these DEGs (*Bcl2l1*, *Hyal2*) were regulated by both CREB1 and NR3C1. CREB1 could be involved in regulating the transcriptional level of three transcription factor genes differentially expressed in the hypothalamus of WAG rats.

Analysis of 111 ISIAH-specific DEGs associated with the oxidative stress response revealed a group of 22 DEGs with CREB1 binding sites in the promoter region. Two of these DEGs (*Fos* and *Jun*) encode transcription factors associated with hypertension. A group of 5 DEGs with NR3C1 binding sites in the promoter region were characterized by statistically insignificant adjusted *p*-values (Table 4).

Under stress, the expression of the *Creb1* gene does not change, and the transcription level of the *Nr3c1* gene significantly decreases in the hypothalamus of both rat strains (Table 5). The presence of a statistically significant correlation between the transcription level of the *Nr3c1* gene and the transcription level of several DEGs associated with the response to oxidative stress and having NR3C1 binding sites in the promoter region (Table 6) suggests that the *Nr3c1* gene may be involved in the regulation of the expression of these genes. In total, in the hypothalamus of ISIAH rats, the transcription level of 89 DEGs out of 170 DEGs associated with the response to oxidative stress correlates with the transcription level of the *Nr3c1* gene (Appendix A). In the hypothalamus of WAG rats, the expression of 20 DEGs out of 59 DEGs associated with the response to oxidative stress correlated with the transcription level of the *Nr3c1* gene (Appendix A).

According to the described results, it can be concluded that under restraint stress, the response to oxidative stress in the hypothalamus of both rat strains can be mediated by the participation of the transcription factors CREB1 and NR3C1. CREB1 can be involved in the regulation of the expression level of the *Fos* gene, as well as several other transcription factor genes, the expression of which correlates with the *Abcc1*, *Nos1*, *Ppargc1a*, *Cyp1b1*, and *Prkaa2*, which most significantly changed the transcription level under stress. It can be assumed that CREB1 and NR3C1 can participate not only in the regulation of the response to oxidative stress but also, possibly, in the regulation of arterial blood pressure under restraint stress, since among the DEGs related to the response to oxidative stress, many genes are associated with hypertension.

## 4. Discussion

The results of the present study show that numerous biological processes affecting important physiological functions are altered in the hypothalamus of hypertensive ISIAH rats and normotensive WAG rats in response to single short-term restraint stress. Some of these processes are common to hypertensive and normotensive animals, while others represent a strain-specific response to stress.

Functional analysis of genes that changed their transcription levels under stress conditions revealed Oxidative phosphorylation as one of the most significantly enriched terms common to both strains. Transcription changes in 74 genes, representing 42.5% of the total list of 174 genes associated with the Oxidative phosphorylation pathway [31], affected links in all five complexes of this pathway (Appendix A). The results of our bioinformatics analysis support the conclusion that activation of oxidative phosphorylation is the most significant process among the metabolic changes in the hypothalamic functioning profile of rats of both strains under the influence of a single short-term restraint stress. This result is consistent with the idea that glucocorticoids may exert effects on mitochondrial transcription and energy metabolism, and with the notion that mitochondria are key components of the stress response, as they are able to orchestrate the adaptive response to stressors, including the adjustment of the bioenergetic, thermogenic, oxidative, and/or apoptotic responses for the re-establishment of homeostasis [38]. Our results suggest a stronger activation of the oxidative phosphorylation pathway by stress in the hypothalamus of hypertensive ISIAH rats compared to WAG.

A large number of DEGs associated with oxidative phosphorylation are also included in the list of DEGs associated with various pathways related to the development of multiple neurodegenerative diseases (NDDs). However, the expression of the genes considered in our study usually decreases with the development of NDDs. The activation of DEGs associated with oxidative phosphorylation observed in our study is most consistent with the changes in their transcription levels presented in the study of the age-related blood–liquor (cerebrospinal fluid) barrier during brain development [39].

One of the most well-known and significant consequences of electron transfer reactions in the process of activation of cellular ATP synthesis through oxidative phosphorylation is the generation of reactive oxygen species (ROS), which can contribute to various biological processes [40,41]. The results of our study suggest that the activation of oxidative phosphorylation can cause a functionally significant increase in oxidative stress in hypothalamic cells, which can trigger processes associated with the response to oxidative stress. The inter-strain differences in oxidative phosphorylation levels found in our study are in good agreement with and explain the significant inter-strain differences observed in the response to oxidative stress between strains.

It has been shown in various animal models that the development of hypertension may be associated with increased production of mitochondrial reactive oxygen species (reviewed in [42]). The production of reactive oxygen species is considered as one of the general molecular and cellular mechanisms that underlie the development of hypertension, since these processes increase neuronal firing in specific brain centers, increase sympathetic outflow, alter vascular tone and morphology, promote sodium retention in the kidney, and, accordingly, increase blood pressure [43]. We have previously shown that the type of stress used in our study statistically significantly increases blood pressure in ISIAH rats, while the increase in BP during stress in control WAG rats does not reach statistical significance [14]. Accordingly, the functional analysis of DEGs identified in our study can help to select the most functionally important hypothalamic genes representing both common and strain-specific links in the regulation of BP in response to oxidative stress in hypertensive ISIAH and normotensive WAG rats.

Analysis of DEGs in the Gene Ontology database revealed a group of 170 genes related to the term “response to oxidative stress”. Statistically significant enrichment of this term was obtained only when analyzing hypothalamic DEGs of hypertensive ISIAH rats. However, 59 DEGs from this list significantly and unidirectionally change the transcription level in the hypothalamus of control WAG rats, which may suggest the activation of ROS production in the hypothalamus of normotensive rats as well. Among the common DEGs that most significantly changed the transcription level during stress, two genes (*Abcc1* and *Nos1*) associated with hypertension deserve the most attention.

*Abcc1* encodes a member of the superfamily of ATP-binding cassette (ABC) transporters. ABCC1 has a major role in blood pressure regulation [44]. It is involved in various physiological functions, including regulation of the level of oxidative stress [45]. It has been shown that ABCC1 is able to influence tissue glucocorticoid sensitivity and modulate negative feedback control of the hypothalamic–pituitary–adrenal (HPA) axis [46,47]. ABCC1 inhibition induces an activation of the HPA axis [46]. Accordingly, it can be assumed that a decrease in the level of *Abcc1* transcription in the rat hypothalamus under conditions of short-term restraint stress may contribute to the activation of the HPA axis in rats of both strains.

*Nos1* encodes a neuronal nitric oxide synthase which is the main producer of NO in the brain. NOS1 is implicated in different physiologically important processes involved in learning and memory development, as well as in synaptic plasticity modulation and neuronal development. A decrease in the level of *Nos1* transcription suggests a decrease in NOS1-derived NO, which may be associated with an increase in oxidative stress and an increase in the inflammatory response (reviewed in [48]). Thus, a decrease in the level of *Nos1* transcription in the hypothalamus of ISIAH and WAG rats under short-term restraint stress may indicate a decrease in the generation and bioavailability of NO in the hypothalamic cells and suggest an increase in oxidative stress and, possibly, may contribute to the development of the inflammatory response.

Both genes (*Abcc1* and *Nos1*) demonstrated similar correlations of their expression with the expression of several transcription factor genes associated with the response to oxidative stress (Table 3). In particular, when analyzing both rat strains, a high correlation of *Abcc1* and *Nos1* expression was shown with the expression of the transcription factor gene *Arnt* associated with hypertension. The protein encoded by *Arnt* is involved in response to dexamethasone. ARNT is also crucial in the response to hypoxia and to hypoglycaemia [49]. ARNT degradation is associated with the induction of the ROS production [50]. Thus, in our study, a decrease in the level of *Arnt* transcription is in good agreement with the assumption of the presence of an increased level of ROS production in the hypothalamus of both rat strains under conditions of limited mobility in a tight wire-mesh cage for 2 h.

Analysis of the promoter regions of common genes associated with the response to oxidative stress allows us to hypothesize that the transcription factors CREB1 and NR3C1 are involved in the regulation of their expression, while the predominant participation of CREB1 in the regulation of ISIAH-specific genes associated with the response to oxidative stress has been shown (Table 4).

CREB1 is known to be activated through phosphorylation by protein kinases and induces gene transcription in response to various extracellular stimuli, including hormonal stimulation [51,52]. CREB1, through control of target gene expression, can affect numerous cellular processes, including cell proliferation, adaptation, survival, differentiation, and others [53]. In a mouse model of DOCA-salt-induced hypertension, it was shown that upon exposure to angiotensin II (AngII), CREB binds to the prorenin receptor promoter, activating it in the hypothalamus [54], which can regulate the activity of the renin–angiotensin system and also affect neuronal function [55]. It has been shown that prorenin activation in the paraventricular nucleus of the hypothalamus causes excitation of the sympathetic nervous system, presumably via the ROS-AP-1-iNOS signaling pathway [56]. In our experiment, *Creb1* transcription levels in the hypothalamus of ISIAH and WAG rats were not altered by stress (Table 5); however, the results of the analysis of the promoter regions of DEGs, as well as the fact that CREB1 activation (phosphorylation) occurs at the post-translational level [57], suggest that the conditions of our experiment may induce CREB1 activation and that this process most likely affects the regulation of the expression of a large number of DEGs involved in the response to oxidative stress.

The involvement of the glucocorticoid receptor in the regulation of the stress response in both rat strains is consistent with the fact that under the stress conditions used, the plasma corticosterone level significantly increases in both ISIAH and WAG rats, regardless of differences in their hypertensive status [14]. NR3C1 (glucocorticoid receptor) is activated by steroid hormones and functions as a transcription factor [58]. It binds to glucocorticoid response elements (GRE) in the promoters of numerous glucocorticoid-sensitive genes, increasing or decreasing their transcription. The involvement of glucocorticoids and their receptors in the response to stress, as well as the involvement of central mechanisms in these processes, is well known [13]. The glucocorticoid receptor is also involved in processes related to metabolism, cell proliferation, and differentiation, as well as inflammatory responses [59,60]. The observed decrease in the *Nr3c1* transcription level in the hypothalamus of both ISIAH and WAG rats can be explained by the fact that glucocorticoid receptors undergo down-regulation following an increase in glucocorticoid levels, which reflects the process of controlling glucocorticoid receptor homeostasis [61]. Downregulation of the *Nr3c1* gene transcription level is also observed with other types of stress, for example, in the hippocampus of male Wistar rats after short-term acute stress during the forced swim test [62].

Both transcription factors (CREB1 and NR3C1) are thought to be involved in the regulation of neuronal plasticity [63]. These transcription factors are functionally related to each other, and their protein–protein interactions have been experimentally confirmed [64], including their induction by cortisol [65]. It has been demonstrated that glucocorticoids can suppress stress-induced CREB phosphorylation in CRH-synthesizing hypothalamic neurons, and it has been suggested that prevention of CREB phosphorylation by glucocorticoids is a possible mechanism for feedback inhibition of CRH (a marker of HPA axis activation) biosynthesis [66].

The stress response in the hypothalamus of hypertensive ISIAH rats compared to normotensive WAG rats has many distinctive features. Among the DEGs that most significantly changed the transcription level only in the hypothalamus of hypertensive ISIAH rats, two genes (*Cyp1b1* and *Fos*) are associated with hypertension.

*Cyp1b1* is a key regulator of cellular redox homeostasis. Its deletion is associated with increased oxidative stress [67]. In our study, the expression of *Cyp1b1*, *Abcc1*, and *Nos1* genes is characterized by similar results of correlation analysis, which suggests similar regulation of changes in transcription levels under stress for these genes. Expression of the *Fos* gene correlates only with the expression of the transcription factor *Jun* (Table 3), which distinguishes this gene from other genes that most significantly changed the transcription level under stress.

Stress-induced activation of *Fos* and *Jun* gene transcription in the hypothalamus of ISIAH rats is consistent with the observation that oxidative stress provokes the expression of these two genes [68], which are subunits of the AP-1 transcriptional complex. It is known that *Fos* gene upregulation is associated with neuronal activation [69], and, accordingly, we can assume that neuronal activation occurs in the hypothalamus of ISIAH rats upon exposure to short-term (2 h) restraint stress. In the hypothalamus of WAG rats, no changes in the transcription level of the *Fos* gene were detected 2 h after the onset of stress. These differences may depend on the genotype and reflect differences in the rate and duration of the stress response, which was previously shown in studies of stress-induced *Fos* gene activation in other strains of hypertensive and normotensive rats [70,71].

Both genes (*Fos* and *Jun*) encode transcription factors associated with hypertension. We have previously shown that *Fos* and *Jun* expression correlates with the expression of the *Npas4* gene (neuronal PAS domain protein 4), the only gene that alternatively changes its transcription level under stress in the hypothalamus of ISIAH and WAG rats. In the hypothalamus of ISIAH rats, *Npas4* expression increases under stress, while it decreases in WAG rats [14]. Considering the above, as well as the fact that the process of activation of *Fos* gene transcription is correlated with the process of stress-induced increase in BP in ISIAH rats [72], we can consider the *Fos* gene as the most likely candidate involved in the process of statistically significant increase in BP under stress in ISIAH rats.

NPAS4 is known to be a neuronal-specific transcription factor that modulates synaptic transmission [73] and is involved in the regulation of excitatory-inhibitory balance within neural circuits [74]. It is known that the processes of the stress-induced depolarization of neuroendocrine cells and the following restoration of resting membrane potential are energy consuming [75]. It has been shown that the induction of NPAS4 in activated cells stimulates energy production by preventing ROS-induced hypoxia-inducible factor 1alpha (HIF1alpha) stabilization, which contributes to maintaining maximal mitochondrial OXPHOS function during excitation of neuroendocrine cells. Alternatively, the knockout of *Npas4* leads to the reduction in OXPHOS activity [76].

According to the results of the KEGG database analysis (Appendix A), the HIF-1 signaling pathway is one of the most significantly enriched terms in the analysis of hypothalamic DEGs in WAG rats, but not in ISIAH rats, suggesting differences in the importance of this metabolic pathway in the hypothalamic response to stress in hypertensive and normotensive rats. Taken together, our results suggest that single short-term restraint stress induces an increase in oxidative phosphorylation, leading to increased ROS production in the rat hypothalamus of both strains. Activation of NPAS4 in ISIAH rats maintains a maximum level of oxidative phosphorylation, preventing the stabilization of HIF1a and, as a result, leading to increased ROS production, thereby inducing oxidative stress, which provokes the expression of AP-1 subunits. In normotensive rats, ROS-induced HIF1a appears to effectively control the level of increase in oxidative phosphorylation through its signaling pathway, thereby preventing high levels of oxidative stress. Thus, the results of our study are in good agreement with the existing literature on the role of oxidative phosphorylation and regulation of the response to oxidative stress under conditions of a single short-term restriction of rat mobility, which can be considered as emotional stress.

## 5. Conclusions

Any organism exists in a constantly changing environment and experiences acute and/or chronic, strong and/or mild stress effects. Various environmental stresses lead to the increased production of reactive oxygen species (ROS), which triggers the cellular response to oxidative stress to restore homeostasis. The regulation of these processes, as our study confirms, largely depends on the genotype.

The interaction between genetic and environmental factors is critical in the development of stress-dependent hypertension. The hypothalamic nuclei are involved in the control of cardiovascular, neuroendocrine, and other physiological functions related to homeostasis [10]. They also regulate sympathetic activity and play a significant role in the development of hypertension [77]. Activation of the sympathetic nervous system due to oxidative stress in the brain mediates an increase in blood pressure in different forms of hypertension, especially in hypertension provoked by stress [78,79].

In the present study, we compared the hypothalamic transcriptomes of adult male hypertensive ISIAH rats, which were a model of stress-sensitive hypertension, and normotensive WAG rats at rest and after exposure to a single short-term (2 h) restraint stress, which can be considered as a kind of emotional stress. The results revealed that oxidative phosphorylation is the most significantly enriched metabolic pathway involved in the response to the stress used in the study. The observed induction of transcription of numerous genes associated with the oxidative phosphorylation metabolic pathway suggests activation of cellular ATP synthesis through oxidative phosphorylation and increased generation of reactive oxygen species. In this regard, in our study, we analyzed in detail a group of DEGs associated with the response to oxidative stress.

However, some important aspects should be noted, which are limitations of our study. One of them is that the results obtained in model animals cannot be directly related to disease pathogenesis in humans. Unfortunately, different forms of hypertension in both humans and hypertensive animal models are in most cases characterized as polygenic diseases, and different sets of genes can be key for different forms of hypertension. The issue of similarities and differences in the molecular genetic mechanisms of development of different forms of hypertension in humans and/or model animals is one of the most pressing. In our previous studies, an attempt was made to compare the genotype of ISIAH rats with the genotype of other hypertensive rat strains. We found that some SNPs are common in several rat strains modeling different forms of hypertension. However, we were unable to find a single SNP common to all hypertensive rat strains [80,81,82]. The search for common links and differences in the metabolomes of blood plasma and urine of ISIAH rats and pharmacological models of hypertension (L-NAME-treated rats with hypertension induced by endothelial dysfunction and rats with hypertension caused by DOCA administration in combination with the salt loading) also reveals both the similarity of some links and numerous differences [83,84].

Another important limitation is that we tested the effects of the stress situation only at one time point (2 h after the onset of stress). It is known that the degree and duration of gene activation may depend not only on the rat’s genotype but also on the type of stress stimulus and the duration of its exposure. Accordingly, the changes in gene expression in the hypothalamus of ISIAH and WAG rats demonstrated in our study correspond to the effect of the used restraint stress for 2 h. With a different duration of stress or exposure to other stress factors, the gene expression profile in the hypothalamus of these rat strains may be partially different.

Since both the mechanisms of the general stress response, independent of the rat strain, and the stress response specific for hypertensive rats are of potential clinical interest, we considered and discussed both aspects of the problem. Knowledge of the molecular mechanisms that shape the response to restraint (emotional) stress will significantly enhance our ability to identify potential molecular targets for further research aimed at developing new therapeutic strategies to prevent the adverse, including pathological, consequences of acute emotional stress.

## Figures and Tables

**Figure 1 antioxidants-13-01302-f001:**
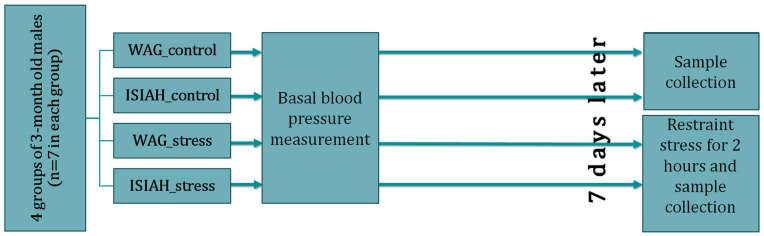
The time course of sample collection.

**Figure 2 antioxidants-13-01302-f002:**
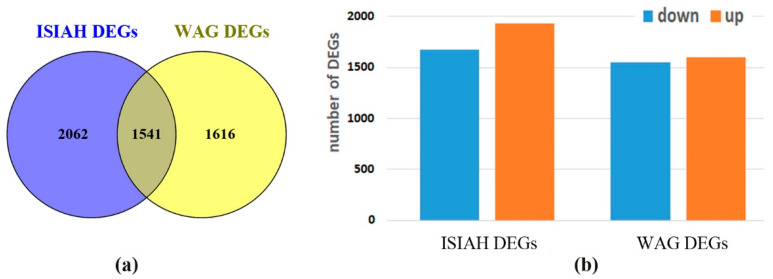
(**a**) Number of identified DEGs in the hypothalamus of ISIAH and WAG rats. (**b**) Changes in gene transcription levels in the hypothalamus of ISIAH and WAG rats when exposed to single restraint stress for 2 h.

**Figure 3 antioxidants-13-01302-f003:**
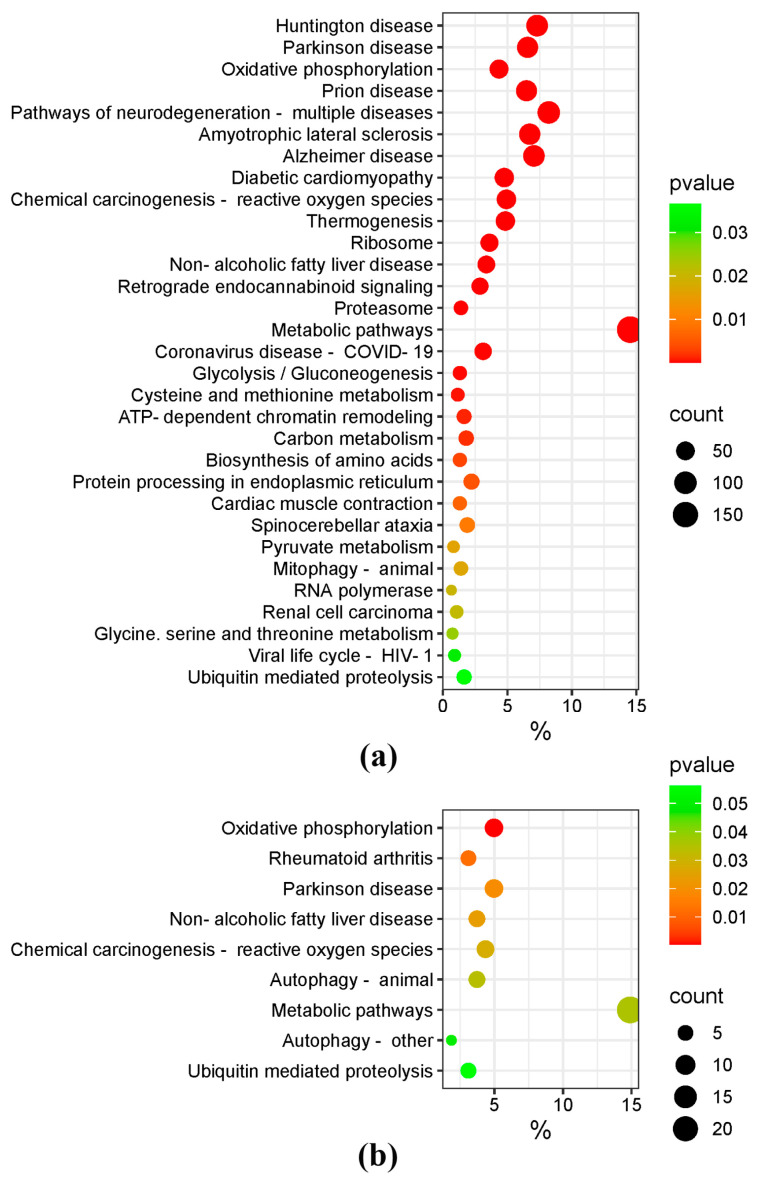
Metabolic pathways for DEGs that contribute most to intergroup differences. (**a**) KEGG analysis for 1219 ISIAH DEGs; (**b**) KEGG analysis for 161 WAG DEGs.

**Figure 4 antioxidants-13-01302-f004:**
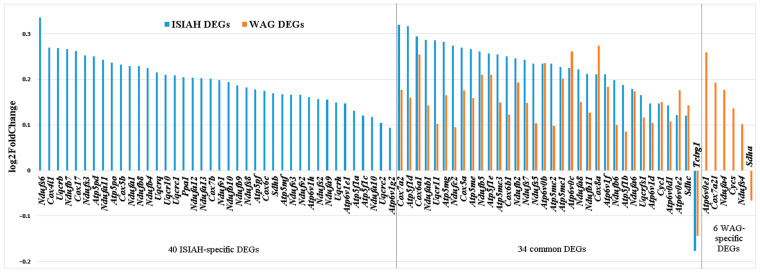
Comparison of hypothalamic ISIAH and WAG DEGs associated with oxidative phosphorylation.

**Figure 5 antioxidants-13-01302-f005:**
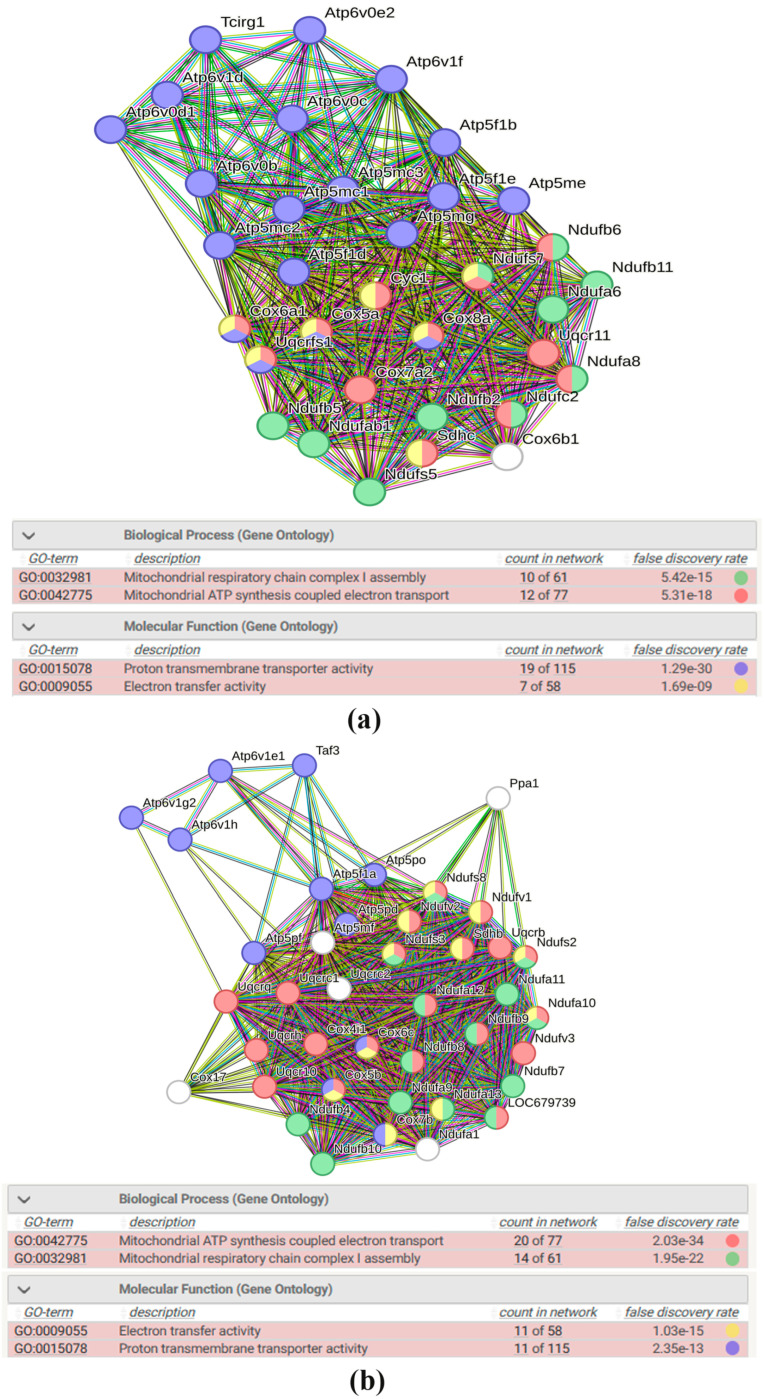
(**a**) Functional annotation of 34 common DEGs associated with oxidative phosphorylation. PPI enrichment *p*-value: <1.0 × 10^−16^. (**b**) Functional annotation of 40 ISIAH-specific DEGs associated with oxidative phosphorylation. PPI enrichment *p*-value: <1.0 × 10^−16^. Purple lines indicate experimentally determined interactions; blue lines denote known interactions from curated databases; dark blue lines represent gene co-occurrence; black lines indicate co-expression; and green lines represent the results of text mining.

**Figure 6 antioxidants-13-01302-f006:**
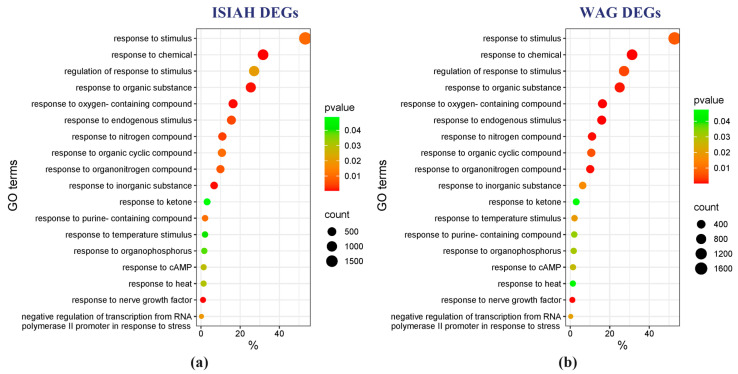
Common GO terms for biological processes, associated with response to stimuli related to restraint stress. (**a**) Analysis of ISIAH DEGs; (**b**) analysis of WAG DEGs.

**Figure 7 antioxidants-13-01302-f007:**
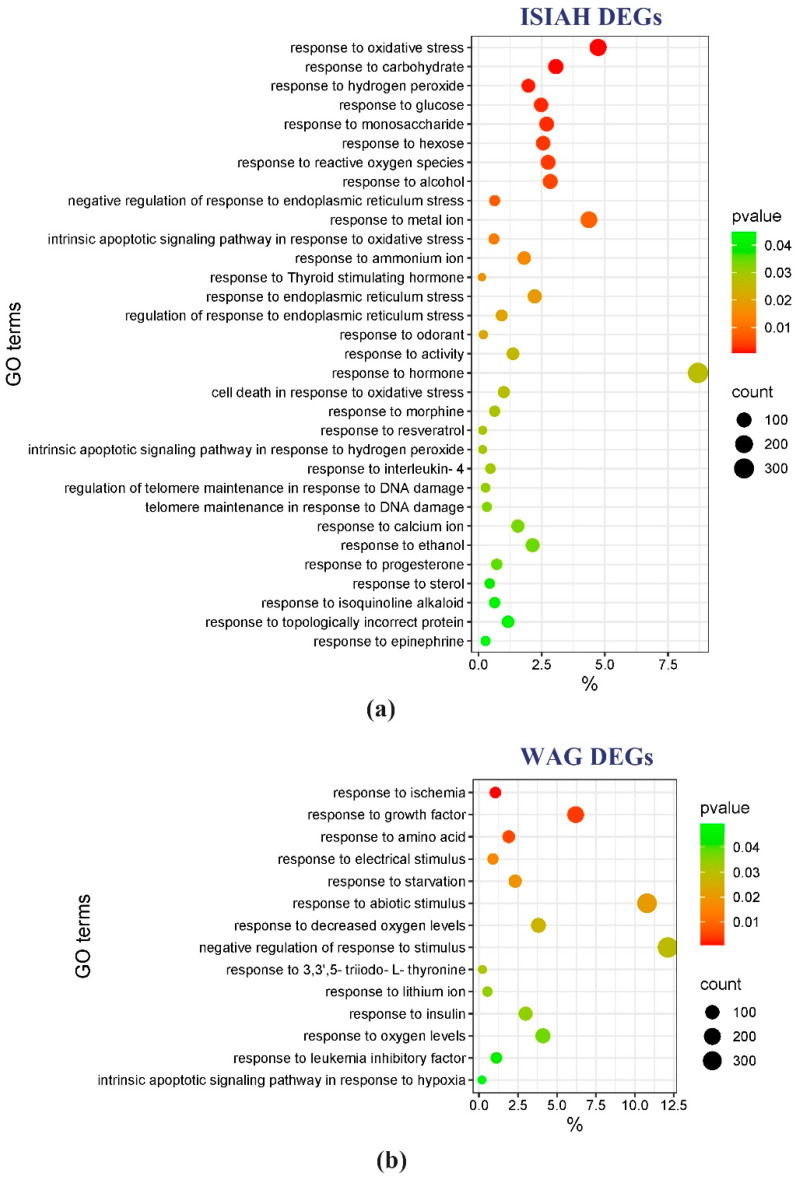
Strain-specific GO terms for biological processes, associated with response to stimuli related to restraint stress. (**a**) Analysis of ISIAH DEGs; (**b**) analysis of WAG DEGs.

**Figure 8 antioxidants-13-01302-f008:**
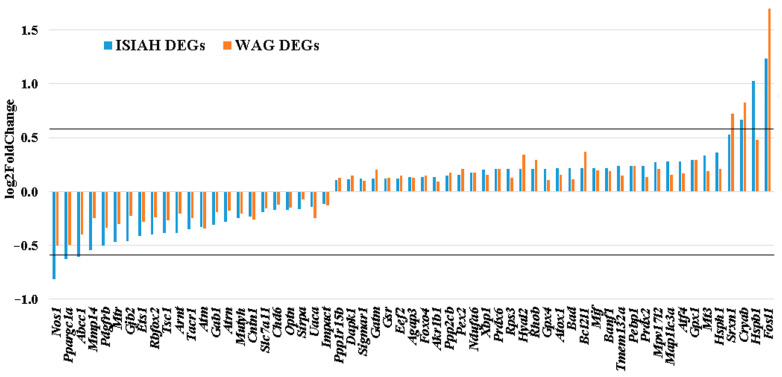
Changes in the transcription levels of common DEGs associated with response to oxidative stress. The horizontal line represents a 1.5-fold change in gene expression level (|log2FoldChange| = 0.585).

**Figure 9 antioxidants-13-01302-f009:**
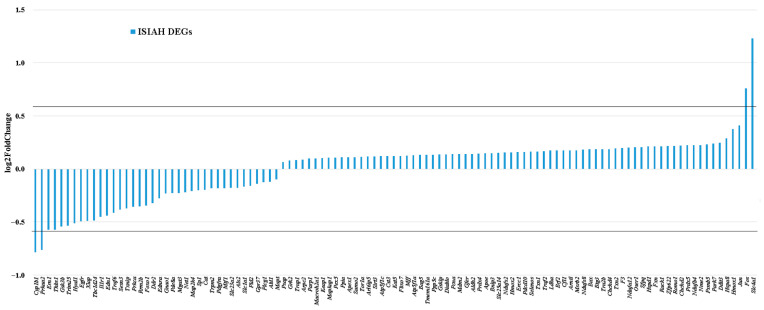
Changes in the transcription levels of ISIAH-specific DEGs associated with response to oxidative stress. The horizontal line represents a 1.5-fold change in gene expression level (|log2FoldChange| = 0.585).

**Figure 10 antioxidants-13-01302-f010:**
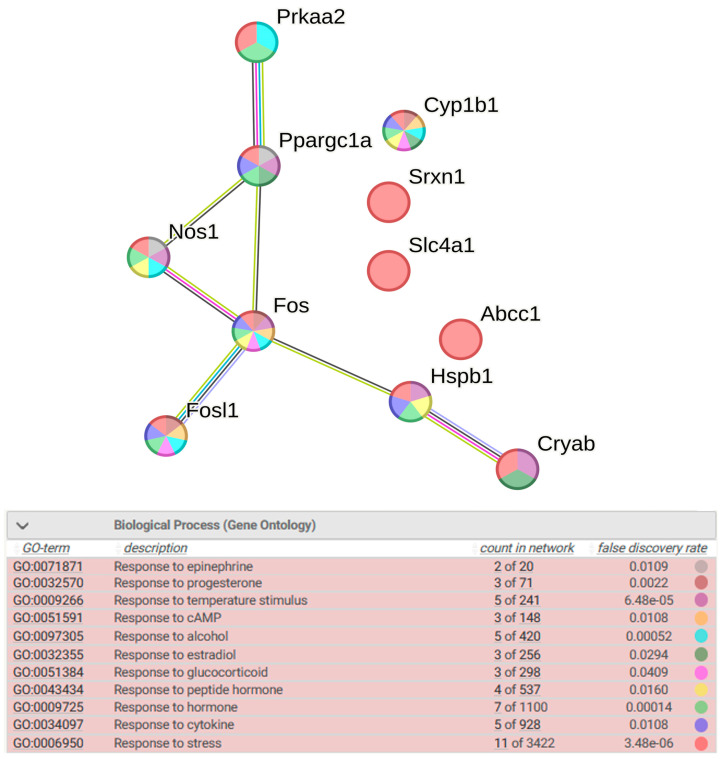
Functional annotation of the 11 most highly altered DEGs associated with response to oxidative stress. Purple lines indicate experimentally determined interactions; blue lines denote known interactions from curated databases; dark blue lines represent gene co-occurrence; black lines indicate co-expression; and green lines represent the results of text mining.

**Figure 11 antioxidants-13-01302-f011:**
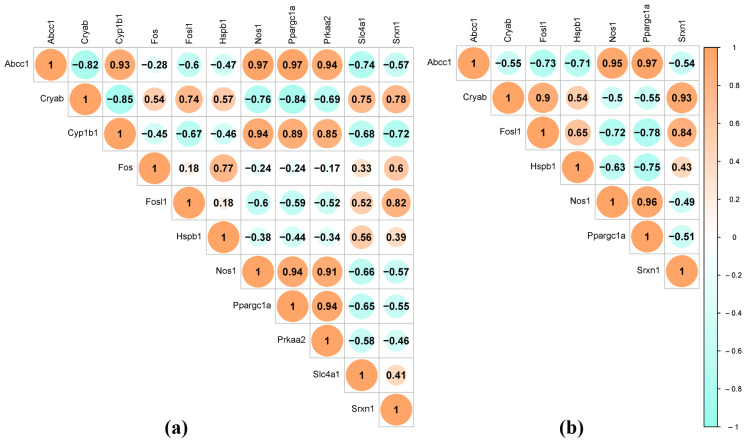
Correlations between the expression of oxidative-stress-response-associated DEGs that showed the most significant changes in transcription level under restraint stress. (**a**) Eleven oxidative-stress-response-associated DEGs in the hypothalamus of ISIAH rats; (**b**) seven oxidative-stress-response-associated DEGs in the hypothalamus of WAG rats. Pearson correlation critical value 0.661 (df = 12, two-tailed *p* value = 0.01). Analysis was performed using SRplot, a free online platform for data visualization and graphing [35].

**Table 1 antioxidants-13-01302-t001:** Oxidative-stress-response-associated DEGs that show the most significant changes in transcription levels upon restraint stress exposure.

Gene Symbol	log2 FC ISIAH_Stress/ ISIAH_Control	*p*adj	log2 FC WAG_Stress/ WAG_Control	*p*adj	Description
*Abcc1 **	**−0.605**	2.97 × 10^−6^	−0.400	1.63 × 10^−7^	ATP binding cassette subfamily C member 1
*Cryab*	**0.670**	2.22 × 10^−19^	**0.824**	4.04 × 10^−63^	crystallin, alpha B
*Fosl1* **#**	**1.233**	7.82 × 10^−4^	**1.699**	2.15 × 10^−6^	FOS like 1, AP-1 transcription factor subunit
*Hspb1*	**1.030**	3.04 × 10^−2^	0.479	2.64 × 10^−5^	heat shock protein family B (small) member 1
*Nos1 **	**−0.816**	1.59 × 10^−2^	−0.499	3.80 × 10^−4^	nitric oxide synthase 1
*Ppargc1a* **#**	**−0.630**	1.31× 10^−4^	−0.496	4.50 × 10^−4^	PPARG coactivator 1 alpha
*Srxn1*	0.530	8.97 × 10^−16^	**0.725**	1.30 × 10^−38^	sulfiredoxin 1
*Cyp1b1 **	**−0.784**	3.50 × 10^−3^			cytochrome P450, family 1, subfamily b, polypeptide 1
*Fos ** **#**	**0.760**	2.94 × 10^−2^			Fos proto-oncogene, AP-1 transcription factor subunit
*Prkaa2*	**−0.762**	1.11 × 10^−2^			protein kinase AMP-activated catalytic subunit alpha 2
*Slc4a1*	**1.231**	2.55 × 10^−2^			solute carrier family 4 member 1 (Diego blood group)

*—genes associated with hypertension. #—genes encoding transcription factors. The values |log2FoldChange| > 0.585 (i.e., the gene transcription level changed more than 1.5 times) are highlighted in bold.

**Table 2 antioxidants-13-01302-t002:** DEGs encoding transcription factors associated with the response to oxidative stress.

Gene Symbol	log2 FC ISIAH_Stress/ ISIAH_Control	*p*adj	log2 FC WAG_Stress/ WAG_Control	*p*adj	Gene ID	Description
*Abl1*	−0.122	3.22 × 10^−2^			311860	ABL proto-oncogene 1, non-receptor tyrosine kinase
*Apex1*	0.112	1.82 × 10^−2^			79116	apurinic/apyrimidinic endodeoxyribonuclease 1
*Arnt **	−0.387	2.72 × 10^−4^	−0.208	1.45 × 10^−2^	25242	aryl hydrocarbon receptor nuclear translocator
*Arntl*	0.177	1.30 × 10^−2^			29657	aryl hydrocarbon receptor nuclear translocator-like
*Atf4 **	0.280	8.24 × 10^−4^	0.171	5.07 × 10^−4^	79255	activating transcription factor 4
*Atm*	−0.332	1.13 × 10^−2^	−0.344	2.59 × 10^−3^	300711	ATM serine/threonine kinase
*Ddit3 **	0.249	1.10 × 10^−2^			29467	DNA-damage inducible transcript 3
*Ets1*	−0.414	4.82 × 10^−9^	−0.280	2.01 × 10^−3^	24356	ETS proto-oncogene 1, transcription factor
*Fbxo7*	0.123	5.63 × 10^−3^			366854	F-box protein 7
*Fos **	0.760	2.94 × 10^−2^			314322	Fos proto-oncogene, AP-1 transcription factor subunit
*Fosl1*	1.233	7.82 × 10^−4^	1.699	2.15 × 10^−6^	25445	FOS like 1, AP-1 transcription factor subunit
*Foxo1 **	−0.347	3.32 × 10^−2^			84482	forkhead box O1
*Foxo4*	0.132	2.07 × 10^−2^	0.151	7.48 × 10^−3^	302415	forkhead box O4
*Jun **	0.410	2.85 × 10^−2^			24516	Jun proto-oncogene, AP-1 transcription factor subunit
*Kat5*	0.121	2.26 × 10^−2^			192218	lysine acetyltransferase 5
*Mtf1*	−0.180	2.71 × 10^−2^			362591	metal-regulatory transcription factor 1
*Nme2*	0.226	8.70 × 10^−5^			83782	NME/NM23 nucleoside diphosphate kinase 2
*Parp1*	0.098	4.11 × 10^−2^			25591	poly (ADP-ribose) polymerase 1
*Ppargc1a*	−0.630	1.31 × 10^−4^	−0.496	4.50 × 10^−4^	83516	PPARG coactivator 1 alpha
*Sfpq*	0.206	1.16 × 10^−3^			252855	splicing factor proline and glutamine rich
*Sirt3*	0.120	1.71 × 10^−2^			293615	sirtuin 3
*Sp1*	−0.199	6.98 × 10^−4^			24790	Sp1 transcription factor
*Trim25*	−0.535	6.89 × 10^−3^			494338	tripartite motif-containing 25
*Xbp1*	0.202	1.71 × 10^−4^	0.154	7.59 × 10^−5^	289754	X-box binding protein 1

* Genes associated with hypertension.

**Table 3 antioxidants-13-01302-t003:** Correlation of DEGs that most significantly change transcription levels under restraint stress with DEGs encoding transcription factors associated with the response to oxidative stress.

**Gene** **Symbol**	**ISIAH DEGs**										
	** *Abcc1 ** **	** *Cryab* **	** *Fosl1* **	** *Hspb1* **	** *Nos1 ** **	** *Ppargc1a* **	** *Srxn1* **	** *Cyp1b1 ** **	** *Fos ** **	** *Prkaa2* **	** *Slc4a1* **
*Abl1*	0.82	−0.48	−0.44	−0.38	0.88	0.66	−0.28	0.73	0.04	0.73	−0.63
*Apex1*	−0.93	0.84	0.53	0.48	−0.89	−0.95	0.57	−0.91	0.23	−0.94	0.64
*Arnt **	0.97	−0.85	−0.56	−0.56	0.95	0.98	−0.54	0.92	−0.23	0.94	−0.68
*Arntl*	−0.05	0.37	0.41	0.26	−0.04	0.01	0.55	−0.30	0.63	−0.05	0.17
*Atf4 *#*	−0.85	0.84	0.59	0.68	−0.84	−0.74	0.71	−0.94	0.58	−0.73	0.68
*Atm #*	0.97	−0.79	−0.47	−0.55	0.94	0.96	−0.42	0.88	−0.19	0.95	−0.69
*Ddit3 *#*	−0.91	0.76	0.41	0.71	−0.90	−0.83	0.51	−0.94	0.47	−0.87	0.67
*Ets1*	0.70	−0.92	−0.75	−0.65	0.60	0.73	−0.85	0.73	−0.52	0.66	−0.76
*Fbxo7*	−0.86	0.90	0.53	0.64	−0.78	−0.92	0.61	−0.87	0.44	−0.86	0.70
*Fos *#*	−0.26	0.61	0.31	0.68	−0.16	−0.26	0.66	−0.46	1.00	−0.20	0.33
*Fosl1*	−0.53	0.72	1.00	0.33	−0.50	−0.53	0.82	−0.60	0.31	−0.48	0.38
*Foxo1 **	0.97	−0.72	−0.48	−0.50	0.97	0.89	−0.42	0.90	−0.18	0.91	−0.68
*Foxo4*	−0.43	0.82	0.70	0.63	−0.29	−0.56	0.77	−0.51	0.68	−0.40	0.38
*Jun *#*	−0.41	0.73	0.36	0.86	−0.28	−0.45	0.63	−0.52	0.88	−0.32	0.52
*Kat5 #*	−0.78	0.85	0.55	0.43	−0.72	−0.87	0.64	−0.81	0.38	−0.78	0.51
*Mtf1 #*	0.95	−0.66	−0.40	−0.56	0.96	0.86	−0.36	0.85	−0.11	0.87	−0.73
*Nme2*	−0.98	0.81	0.57	0.60	−0.97	−0.95	0.51	−0.91	0.19	−0.92	0.70
*Parp1*	−0.18	0.56	0.39	0.28	−0.06	−0.32	0.61	−0.30	0.58	−0.18	0.17
*Ppargc1a*	0.95	−0.86	−0.53	−0.63	0.89	1.00	−0.53	0.88	−0.26	0.95	−0.67
*Sfpq*	−0.73	0.78	0.62	0.63	−0.71	−0.62	0.77	−0.78	0.50	−0.58	0.75
*Sirt3*	−0.86	0.88	0.54	0.58	−0.78	−0.92	0.58	−0.84	0.36	−0.86	0.64
*Sp1 #*	0.85	−0.88	−0.63	−0.53	0.77	0.91	−0.61	0.79	−0.28	0.82	−0.75
*Trim25*	0.95	−0.71	−0.52	−0.43	0.97	0.91	−0.44	0.88	−0.05	0.92	−0.64
*Xbp1 #*	−0.79	0.92	0.54	0.76	−0.67	−0.87	0.68	−0.76	0.52	−0.80	0.79
**Gene Symbol**	**WAG DEGs**										
	** *Abcc1 ** **	** *Cryab* **	** *Fosl1* **	** *Hspb1* **	** *Nos1 ** **	** *Ppargc1a* **	** *Srxn1* **				
*Arnt **	0.92	−0.49	−0.62	−0.60	0.91	0.84	−0.50				
*Atf4 *#*	−0.68	0.75	0.82	0.48	−0.66	−0.68	0.68				
*Atm #*	0.93	−0.58	−0.76	−0.62	0.92	0.87	−0.62				
*Ets1*	0.04	−0.63	−0.49	−0.05	−0.05	−0.11	−0.73				
*Fosl1*	−0.70	0.88	1.00	0.61	−0.69	−0.68	0.86				
*Foxo4*	−0.15	0.71	0.51	0.06	−0.09	−0.14	0.71				
*Ppargc1a*	0.96	−0.49	−0.68	−0.77	0.96	1.00	−0.46				
*Xbp1 #*	−0.67	0.83	0.83	0.72	−0.60	−0.64	0.70				

* Genes associated with hypertension. #—TFs that can be regulated by CREB1. Pearson correlation (r) critical value 0.661 (df = 12, two-tailed *p* value = 0.01). r > |0.9| are shown in red; r > |0.8| are shown in brown; r > |0.7| are shown in yellow.

**Table 4 antioxidants-13-01302-t004:** Enrichment analysis of the promoter regions of DEGs associated with the response to oxidative stress.

Enrichr Terms	Number of DEGs	Adjusted *p*-Value	DEGs Associated with the Response to Oxidative Stress
Analysis of 170 DEGs			
CREB1 23762244 ChIP-Seq HIPPOCAMPUS Rat	38	4.52 × 10^−5^	*Atf4 *#*, *Atm #*, *Bad **, *Bcl2l1 **, *Brf2*, *Cfl1 **, *Chd6*, *Ddit3 *#*, *Diablo*, *Eef2 **, *Fos *#*, *Gpx4 **, *Gsk3b **, *Hspa8 **, *Hsph1*, *Hyal2 **, *Jun*#*, *Kat5 #*, *Ldha **, *Map1lc3a **, *Mif **, *Mmp14*, *Mtf1 #*, *Ndufs8*, *Pcna **, *Pde8a*, *Ppia **, *Psmb5*, *Rhob*, *Rps3*, *Sesn3*, *Srxn1*, *Sp1 #*, *Tra2b*, *Traf2*, *Txn2*, *Txnip **, *Xbp1 #*
NR3C1 34362910 ChIP-Seq WistarRat Hippocampus Stress	12	4.41 × 10^−2^	*Bcl2l1 **, *Cryab*, *Egfr **, *F3 **, *Gab1*, *Hyal2 **, *Mtr **, *Pebp1*, *Prkca **, *Slc7a11*, *Trpm2*, *Txnip**
Analysis of 59 DEGs common for ISIAH and WAG			
CREB1 23762244 ChIP-Seq HIPPOCAMPUS Rat	16	6.14 × 10^−3^	*Atf4 *#*, *Atm #*, *Bad **, *Bcl2l1 **, *Chd6*, *Eef2 **, *Gpx4 **, *Hsph1*, *Hyal2 **, *Map1lc3a **, *Mif **, *Mmp14*, *Rhob*, *Rps3*, *Srxn1*, *Xbp1 #*
NR3C1 34362910 ChIP-Seq WistarRat Hippocampus Stress	7	4.53 × 10^−2^	*Bcl2l1 **, *Cryab*, *Gab1*, *Hyal2 **, *Mtr **, *Pebp1*, *Slc7a11*
Analysis of 111 DEGs (ISIAH-specific response to oxidative stress)			
CREB1 23762244 ChIP-Seq HIPPOCAMPUS Rat	22	1.20 × 10^−2^	*Brf2*, *Cfl1 **, *Ddit3 *#*, *Diablo*, *Fos *#*, *Gsk3b **, *Hspa8 **, *Jun *#*, *Kat5 #*, *Ldha **, *Mtf1 #*, *Ndufs8*, *Pcna **, *Pde8a*, *Ppia **, *Psmb5*, *Sesn3*, *Sp1 #*, *Tra2b*, *Traf2*, *Txn2*, *Txnip **
NR3C1 34362910 ChIP-Seq WistarRat Hippocampus Stress	5	4.75 × 10^−1^	*Egfr **, *F3 **, *Prkca **, *Trpm2*, *Txnip **

*—genes are associated with hypertension; #—transcription factor genes.

**Table 5 antioxidants-13-01302-t005:** Changes in the transcription level of the *Nr3c1* and *Creb1* genes in the hypothalamus of ISIAH and WAG rats under restraint stress.

Gene Symbol	Gene ID	log2 FCISIAH Stress\ ISIAH Control	log2 FCWAG Stress\ WAG Control	Description
*Creb1 **	81646	−0.180	0.020	cAMP responsive element binding protein 1
*Nr3c1 **	24413	−0.133 ^#^	−0.113 ^#^	nuclear receptor subfamily 3, group C, member 1

*—genes are associated with hypertension. ^#^ *p*adj < 0.05.

**Table 6 antioxidants-13-01302-t006:** Correlation of the transcription level of the *Nr3c1* gene with the transcription level of genes associated with the response to oxidative stress and having NR3C1 binding sites in the promoter region.

ISIAH DEGs		WAG DEGs	
Gene_Symbol	r	Gene_Symbol	r
*Slc7a11*	**0.925**	*Gab1*	**0.788**
*Pebp1*	−**0.909**	*Mtr*	0.641
*Hyal2*	−**0.872**	*Hyal2*	−0.622
*Bcl2l1*	−**0.872**	*Pebp1*	−0.568
*Mtr*	**0.805**	*Cryab*	−0.542
*Cryab*	−**0.800**	*Bcl2l1*	−0.484
*Egfr*	**0.734**	*Slc7a11*	−0.286
*Prkca*	0.643		
*Gab1*	0.420		
*Txnip*	0.391		
*Trpm2*	0.279		
*F3*	−0.267		

Statistically significant correlations are shown in bold (df = 12; Pearson correlation critical value 0.661; two-tailed *p* value = 0.01).

## Data Availability

All relevant data are available in Appendix A.

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
