# Peer review of "Effect of Short-Term Restraint Stress on the Expression of Genes Associated with the Response to Oxidative Stress in the Hypothalamus of Hypertensive ISIAH and Normotensive WAG Rats"

_antioxidants, 2024, doi:10.3390/antiox13111302_

Round 1
Reviewer 1 Report
This article investigates the role of oxidative stress and chronic inflammation in the pathogenesis of arterial hypertension, highlighting the involvement of oxidative stress in the central nervous system. The research compares gene expression profiles between hypertensive ISIAH rats and normotensive WAG rats, revealing significant differences in genes linked to blood pressure regulation and central nervous system function. The study identifies common and strain-specific gene responses to short-term restraint stress, particularly emphasising genes related to ion transport and signaling. The authors state that these findings provide insights into potential molecular targets for hypertension treatment.
It is an elegant study but with limitations with potential built in biases and over extrapolated conclusions.
The ISIAH rats have an inherited stress-induced arterial hypertension, so this study is demonstrating a possible action of this mutation rather than a general response to stress. In human hypertension it is chronic stress that is thought to be a causative factor not short term stress. Chronic stress involves different physiological and molecular adaptions which might not be captured by the studies design.
The control group, the WAG rats do not have this genetic feature and therefore their response could relate to their different genetic make up. Therefore, it is difficult to extrapolate these findings to human hypertension and treatment. Hypertension also has a multifactorial causal basis of which these findings may or may not be associated. Further human research would be required to validate these findings before therapeutic considerations can be made.
So, although the study provides important insights into the genetic response short term stress in genetically specific hypertensive rats and normotensive rats, caution should be exercised when extrapolating these findings to humans.
Author Response
Response to Reviewer 1 Comments
The authors are grateful to the reviewer for carefully reading the manuscript and making comments to improve the text. Below are the detailed responses and links to the lines of the text containing the corresponding corrections. Corrections made in the text are highlighted in red.
Major comments
This article investigates the role of oxidative stress and chronic inflammation in the pathogenesis of arterial hypertension, highlighting the involvement of oxidative stress in the central nervous system. The research compares gene expression profiles between hypertensive ISIAH rats and normotensive WAG rats, revealing significant differences in genes linked to blood pressure regulation and central nervous system function. The study identifies common and strain-specific gene responses to short-term restraint stress, particularly emphasizing genes related to ion transport and signaling. The authors state that these findings provide insights into potential molecular targets for hypertension treatment.
It is an elegant study but with limitations with potential built in biases and over extrapolated conclusions.
Detail comments
The ISIAH rats have an inherited stress-induced arterial hypertension, so this study is demonstrating a possible action of this mutation rather than a general response to stress. In human hypertension it is chronic stress that is thought to be a causative factor not short term stress. Chronic stress involves different physiological and molecular adaptions which might not be captured by the studies design.
The control group, the WAG rats do not have this genetic feature and therefore their response could relate to their different genetic make up. Therefore, it is difficult to extrapolate these findings to human hypertension and treatment. Hypertension also has a multifactorial causal basis of which these findings may or may not be associated. Further human research would be required to validate these findings before therapeutic considerations can be made.
So, although the study provides important insights into the genetic response short term stress in genetically specific hypertensive rats and normotensive rats, caution should be exercised when extrapolating these findings to humans.
Answer: The authors agree with the reviewer's comment that the findings presented in the manuscript cannot be directly extrapolated to human hypertension and treatment. The text of the abstract has been amended to avoid over extrapolated conclusions (lines 33-34). A description of the limitations of the study has also been added to the Conclusion section (lines 655-678 and 683).

Reviewer 2 Report
The authors reveal comprehensive genetic changes in the hypothalamus during stress-responsive hypertension in hypertensive organisms. I find the data in this paper interesting and innovative. Please answer my questions and consider revising the paper if necessary.
1. You mentioned that you measured blood pressure in both ISIAH and WAG rat groups. What are the specific blood pressure values (and time course, if possible) in this study?
2. Did the characteristics of the ISIAH rats or hypertension itself have a significant impact on the results of this study?
3. What kind of clinicopathological model does the ISIAH rat represent? Is ISIAH an animal model of white coat hypertension? If so, would you expect the results to be substantially different in a general hypertension model such as salt-sensitive hypertension?
4. Since Fos expression increases in various nuclei in the brain during acute stress, it is expected to increase in WAG as well. Why do you think it increases only in ISIAH?
5. In this study, the authors are studying genetic changes in the hypothalamus due to acute stress. Would the experimental results be different if examined under chronic stress?
6. As the authors know, different nuclei in the hypothalamus have different functions. Depending on where Fos appears, there are various possibilities, such as whether it affected adrenal function, or whether it affected the autonomic nervous system and the production and secretion of hormones produced in the hypothalamus (e.g., vasopressin). Is it possible to investigate RNA sequencing limited to specific regions in the hypothalamus and perform detailed verification?
7. The authors used an anesthetic to eliminate the effect of blood pressure. How do the authors expect this anesthetic to affect emotional responses and gene expression?
1. How did you identify the hypothalamus?
2. Are there differences in the expression levels (e.g., blood concentrations) of glucocorticoids, NO, etc. between ISIAH and WAG rats?
Author Response
Response to Reviewer 2 Comments
The authors are grateful to the reviewer for carefully reading the manuscript and making comments to improve the text. Below are the detailed responses and links to the lines of the text containing the corresponding corrections. Corrections made in the text are highlighted in red.
Major comments
The authors reveal comprehensive genetic changes in the hypothalamus during stress-responsive hypertension in hypertensive organisms. I find the data in this paper interesting and innovative. Please answer my questions and consider revising the paper if necessary.
- You mentioned that you measured blood pressure in both ISIAH and WAG rat groups. What are the specific blood pressure values (and time course, if possible) in this study?
Answer: In section “2.1. Animals”, a reference to our previous publication characterizing both changes in blood pressure and plasma corticosterone concentration in the experimental rats used has been added. Text with a brief description of the measurement of plasma corticosterone concentration has also been added (lines 104–107).
- Did the characteristics of the ISIAH rats or hypertension itself have a significant impact on the results of this study?
Answer: Thank you for the leading question. Indeed, the article is devoted to differences in the stress response in the hypothalamus of hypertensive and normotensive rats, but the final sentence of the abstract does not emphasize this. We have corrected the text of the final part of the abstract (lines 29-32), as well as in the formulation of the study objective (line 75), to emphasize the presence of a strain-specific impact of the hypertensive state of ISIAH rats.
- What kind of clinicopathological model does the ISIAH rat represent? Is ISIAH an animal model of white coat hypertension? If so, would you expect the results to be substantially different in a general hypertension model such as salt-sensitive hypertension?
Answer: Yes, we believe that hypertension in ISIAH rats most closely corresponds to white coat hypertension in humans.However, it is difficult to talk about the similarity of pathogenesis of these two forms of hypertension.And the reason is not only that the results obtained in model animals cannot be directly related to pathogenesis in humans.Unfortunately, different forms of hypertension in both humans and hypertensive animal models are in most cases characterized as polygenic diseases, and different sets of genes can be key for different forms of hypertension.The issue of similarities and differences in the molecular genetic mechanisms of development of different forms of hypertension in humans and/or model animals is one of the most pressing.In our previous studies, an attempt was made to compare the genotype of ISIAH rats with the genotype of other hypertensive rat strains.We found that some SNPs are common in several rat strains modeling different forms of hypertension.However, we were unable to find a single SNP common to all hypertensive rat strains (Ershov at al., 2017; Devyatkin et al., 2020).The search for common links and differences in the metabolomes of blood plasma and urine of ISIAH rats and pharmacological models of hypertension (L-NAME-treated rats with hypertension induced by endothelial dysfunction and rats with hypertension caused by DOCA administration in combination with the salt loading) also reveals both the similarity of some links and numerous differences (Seryapina et al., 2024; Sorokoumova et al., 2024).
Understanding the importance of studying both the general mechanisms of response to stress and the specific links in the manifestation of pathology in normotensive and hypertensive rats, we tried to reflect in detail in this study both the general and strain-specific mechanisms of response to stress in hypertensive and normotensive rats.
We have added a description of this limitation in the Conclusion section (Lines 656-670).
Ershov, N.I.; Markel, A.L.; Redina, O.E. Strain-Specific Single-Nucleotide Polymorphisms in Hypertensive ISIAH Rats. Biochemistry (Mosc). 2017, 82, 224-235, doi:10.1134/S0006297917020146.
Devyatkin, V.A.; Redina, O.E.; Muraleva, N.A.; Kolosova, N.G. Single-Nucleotide Polymorphisms (SNPs) Both Associated with Hypertension and Contributing to Accelerated-Senescence Traits in OXYS Rats. Int J Mol Sci 2020, 21, doi:10.3390/ijms21103542.
Seryapina, A.A.; Malyavko, A.A.; Polityko, Y.K.; Yanshole, L.V.; Tsentalovich, Y.P.; Markel, A.L. Metabolic profile of blood serum in experimental arterial hypertension. Vavilovskii Zhurnal Genet Selektsii 2023, 27, 530-538, doi:10.18699/VJGB-23-64.
Sorokoumova, A.A.; Seryapina, A.A.; Polityko, Y.K.; Yanshole, L.V.; Tsentalovich, Y.P.; Gilinsky capital Em, C.A.C.; Markel capital A, C. Urine metabolic profile in rats with arterial hypertension of different genesis. Vavilovskii Zhurnal Genet Selektsii 2024, 28, 299-307, doi:10.18699/vjgb-24-34.
- Since Fos expression increases in various nuclei in the brain during acute stress, it is expected to increase in WAG as well. Why do you think it increases only in ISIAH?
Answer: The authors are grateful to the reviewer for drawing attention to the inaccuracy of the wording in the text describing this part of the results. We have made the necessary corrections to the discussion text (lines 591–597) and have added the appropriate text reflecting the limitation of our study in the conclusion section (lines 671–678).
- In this study, the authors are studying genetic changes in the hypothalamus due to acute stress. Would the experimental results be different if examined under chronic stress?
Answer: The answer to this question is reflected in the text of our study's limitations, included in the conclusion section (lines 671–678).
- As the authors know, different nuclei in the hypothalamus have different functions. Depending on where Fos appears, there are various possibilities, such as whether it affected adrenal function, or whether it affected the autonomic nervous system and the production and secretion of hormones produced in the hypothalamus (e.g., vasopressin). Is it possible to investigate RNA sequencing limited to specific regions in the hypothalamus and perform detailed verification?
Answer: This comment sounds like a potential interesting experimental design. However, the reviewer's proposed further research requires dedicated funding for the study.
- The authors used an anesthetic to eliminate the effect of blood pressure. How do the authors expect this anesthetic to affect emotional responses and gene expression?
Answer: Hypothalamic samples were collected 7 days after basal BP measurement. This period of time is sufficient for the ether anesthesia-induced increase in corticosterone and the associated change in gene levels to return to pre-stress levels. The authors agree with the reviewer that the time courses of blood pressure testing in rats were not presented clearly in the text of the manuscript. The text in the Materials and Methods section has been corrected and a figure has been added to clarify the time course of the experiment (lines 91, 94, 96, Figure 1).
Detail comments
- How did you identify the hypothalamus?
Answer: The hypothalamus was isolated according to known anatomical location. Paxinos G, Watson C. (1998). The rat brain in stereotaxic coordinates, Ed 4. San Diego, CA: Academic Press. A reference has been added to the text of the Materials and Methods section (see line 101).
- Are there differences in the expression levels (e.g., blood concentrations) of glucocorticoids, NO, etc. between ISIAH and WAG rats?
Answer: Section “2.1. Animals” has been updated with a text describing the measurement of corticosterone concentration and a reference to our previous publication characterizing both changes in blood pressure and plasma corticosterone concentration in the experimental rats used (lines 104–107).

Round 2
Reviewer 1 Report
There are still limitations got this study but the authors have addressed this and madde the paper stronger and more relatable. It now adds a significant body of work to the scientific community.
The results are still a little overwhelming but necessary to show the range of investigations carried out.
This is a far better paper with the additional work.
Reviewer 2 Report
I believe the authors responded appropriately to my comments and improved the quality of their paper by revising it.
I believe the authors responded appropriately to my comments and improved the quality of their paper by revising it.